# Designing and Testing Cold-Formed Rounded Connections Made on a Prototype Station

**DOI:** 10.3390/ma12071061

**Published:** 2019-03-31

**Authors:** Michał Rejek, Nikodem Wróbel, Jolanta Królczyk, Grzegorz Królczyk

**Affiliations:** 1Mechanical Design Department, INTROL PRO-ZAP Sp. z o.o., ul. Grabowska 47a, 63-400 Ostrów Wielkopolski, Poland; michal.rejek@gmail.com (M.R.); nikodem.wrobel@gmail.com (N.W.); 2Faculty of Mechanical Engineering, Opole University of Technology, ul. St. Mikołajczyka 5, 45-001 Opole, Poland; g.krolczyk@po.opole.pl

**Keywords:** prototype station, fixed joints, inseparable tight joints, formed joints, bent joints, cold-formed joints, clinched joint, clinching, joint strength, joint characterization and testing

## Abstract

This paper presents the design of cold-formed rounded connections between a tube and a connecting block and the analysis of test results that were carried out with six fabricated samples. The joints manufactured on a specially designed prototype station were made by forming tools that were adjusted to connecting elements regarding the diameter and the shape. All of the samples prepared for this study were of the same diameters relating to the diameter of a hole in a connecting block and the outer diameter of a pipe flange. However, they were different concerning the height of the connecting block flange. The article presents features of joints that were manufactured with a designed forming tool on the prototype station. The achieved connections were examined in destructive testing (Micrography, Tensile Strength Test) and in non-destructive testing (Leakage Test). The research project aims were to state the differences in energy consumption of made connections and extend the concept of cold-formed rounded connections. Furthermore, this article presents the effects of FEA simulation of the cold-formed joint based on the results of destructive and non-destructive tests.

## 1. Introduction

Connections that formed on account of material deformation in joining elements can be often found in production of components in different industries, i.e., in the case of the automotive industry, they can be found in components of fuel pumps, in all kinds of coolers and condensers, or even in large components, such as engine blocks. In order to perform the connections, mechanical clinching and crimping processes are undertaken. Subsequently, two elements are geometrically combined with partial deformation being caused by a punch and a die [1,2]. The pipe end forming process studied in the article assumes the use of clamps to hold the formed material and the forming tools in order to significantly change the shape of pipe end. In such operations, a pipe end usually needs to be adjusted to the other element. The machine that is used in the end forming process is most frequently a ram-type end former, which can perform different end pipe types, e.g., a bead, a spherical closed end, and a shoulder expansion. The pipe ends that are formed by a machine of such a type can be divided into two groups: with a reduced outer diameter of pipe and with an expanded outer diameter of pipe [3]. Regardless of the final application, each formed joint needs to meet certain requirements. It is often difficult due to working conditions or external factors, such as mechanical stress, thermal stress (high or low ambient temperature), high pressure of medium flowing through the formed joint, substantial difference between the medium and the ambient temperature, and features of the substance flowing through the joint (the substance can be aggressive) [4,5].

All of the conditions mentioned above, as well as working conditions, must be taken into account during the process of designing the joint. However, the economic aspect of the issue should be also considered. Even the best designed connections that neglect economic assumptions will not be implemented. Appropriate economical approach to the joint production can be revealed in various aspects, such as the appropriate selection of material in relation to production processes, working parameters, or choosing an appropriate manufacturing method, e.g., the joint can be welded, bonded using adhesives, crimped mechanically, clinched, formed [6].

The three last methods listed have significant advantages, i.e., no influence on the joint critical surfaces or no high temperature effects on elements [7]. An additional difficulty that can be encountered in the design of joints for the automotive industry is the limited space available for forming tools, which must withstand the high forces needed to cause the permanent deformation of connecting elements [8].

In order to achieve satisfying quality of joints with high specific parameters conducting a process of high repeatability, numerous researchers investigated the process of forming joints. Kaščák et al. [2] investigated a non-cutting conventional mechanical clinching to form rounded joints with a rigid die while using two types of materials-ferrous and non-ferrous. They applied different material combinations of steel and aluminum alloy sheets for clinching. The investigation shows, among others, that to achieve better results regarding the load-bearing capacity of mechanically clinched joints, there should be used arrangements: a steel sheet on the punch side and an aluminum alloy sheet on the die side. Mucha [9] conducted an investigation into the lock forming mechanism in clinching joints with the use of high-strength steel H320LA. The joints were investigated while using the finite element analysis. Researcher established that the die groove width is the most important parameter affecting the material flow and energy consumption in the joining process. Mucha et al. [10] continued analyzing the clinching process, especially the relationship between the strength of the joint and its bottom thickness. Reducing the embossment thickness increases the joint strength; however, it is connected with increasing the forming force. Coppieters et al. [11] studied the bottom defects that were generated while creating the permanent interlock in the case of clinching with a rigid die. The study shows that the observed bottom cracks can be predicted by means of the axisymmetric FE model while using the modified Rousselier model. Chen et al. [7] looked into two different compressing methods of forming clinched joints: clinching with an extensible die combined with the compressing process that was carried out with a pair of flat dies and clinching with an extensible die combined with the compressing process that were carried out with a flat die and a bumped die. They used different sheets of aluminum alloy. The forming processes that were studied in the article can be used to get lower protrusion of formed joints and neat appearance. The researchers established that both methods are efficient and reliable, and they can increase the cross-tensile strength and the tension-shearing strength of the clinched joint. On the one hand, the process of compressing with a flat die and a bumped die delivers better results regarding the increase of the joint strength. On the other hand, smaller protrusion can be achieved while using the process with a pair of flat dies. As for the different material combinations, the aluminum alloy 6061 that was chosen as the material for the upper sheet instead of the aluminum alloy 5052 sheet provides higher cross-tensile and tension-shearing strength. Chen et al. [12] discussed a mechanical clinching method with a flat die and with different material flow than in a normal clinching method. The material of upper sheets flows radially and upward, while the material of lower sheets flows radially and downward. Thanks to a flat anvil, there is no exterior protrusion in case of the lower sheet. Chen et al. [13] continued investigating the Flat Clinching Process with the use of experimental and numerical methods. The results were compared to the experimentally formed joints and further development of the process was investigated. The scientists studied the influence of punch diameter and holder force on behavior of deforming materials. According to the obtained data, a punch with a larger diameter could give larger neck thickness and interlocking length, which affects the quality of the clinched connections. Mostafa et al. [14] studied the clinching process that was carried out with a flat die and various holder shapes using aluminum alloy sheets in different temper states. The researchers established that the forming ability is affected by the temper state of aluminum alloy. Using annealed materials to form joints results in better quality of joints in relation with bigger interlock depth. The groove that is designed in the blank holder also presents high influence on a formed joint. The depth of the groove had a significant impact on the interlock formation, and then the diameter of the groove had impact on the reduction in the neck thickness. Studies of clinched joints revealed that the neck thickness and the interlock depth strongly influence shear strength of clinched joints and that the peel strength of formed joints highly depends on the neck thickness. Mostafa et al. [15] continued analyzing the clinching process with a flat die using the FE Analysis. The obtained data was compared to the data received in previous experiments. The researchers settled and confirmed that strength of clinched joints can be increased by reduction of the blank holder depth, which allows for controlling the material flowing in the radial direction, instead of the axial direction. It makes the interlock formation bigger. The next parameters that influence interlock forms are the rounded edges of the blank holder or the groove diameter of the stepped edge blank holders. Increasing any of the parameters increases the interlock depth. Mucha [16] analyzed how the material properties and joining process parameters affect the behavior of self-piercing riveting joints that were made with a solid rivet. According to the study, the rivet material significantly affects the joint strength in the case of conventional riveted joints, and the lowest die and punch load force are the crucial factors related to the tool lifetime. When two sheets of different thickness are used in the process, the thicker sheet should be used as the lower sheet to improve the joint strength. He et al. [17] investigated the joining process, called Self-Pierce Riveting (SPR), which is based on inserting a rivet in two materials, thus combining them together. Authors discussed the mechanism of forming the joint when considering the types of defects that occurred, the mechanical strength, and corrosion properties of manufactured joints. Furthermore, researchers studied the results of FE analysis in order to determine the variables that affect SPR properties. Han et al. [18] compared two methods of joining elements: SPR and RSW (Resistance Spot Welding). Despite various parameters of the processes, which being optimized can significantly influence the results, researchers concluded, among others, that the joints obtained from the results of SPR have a tendency to be less various than the RSW samples. Moreover, elements combined with the use of Self-Pierce Riveting technology tend to have similar or higher peel strength than the parts that were manufactured with the use of the Resistance Spot Welding method. Briskham et al. [19] compared the SPR method with Spot Friction Joining and Resistance Spot Welding. Authors established that, in comparison with the two other methods, the advantages of the SPR method are, among others, the best mechanical properties and the possibility to join elements that are made of different materials. On the other hand, the biggest economic issue of SPR connection is the rivet, which significantly affects the cost of the joints. The second disadvantage of the process is the complexity of changes related to the joint configuration, which is connected with the replacement of a rivet and a die. Additionally, the bottom element has to be sufficiently thick, otherwise the rivet will not form a strong interlock. Kaščák et al. [20] investigated the joining process, called ClinchRivet. Researchers focused on the testing process using aluminum rivets rather than steel ones. Their results, achieved while testing the formed joints, were satisfying, so aluminum rivets can be successfully used in the ClinchRivet process to join the tested materials. Due to the many advantages of the process, ClinchRivet technology can be frequently used in massive production.

Another way used to form joints, which is different than the previously described methods, is the crimping method that is used in production of cable connectors. Research on this type of connections has been carried out for a long time, e.g., in 1967, Hayner [21] studied connection production depending on settings of the forming tool. Subsequently, Mocellini and Petitprez [22] presented studies of the crimping process used to connect a multi-strand wire to a tube component. In their elaboration, there are shown results of formed connections made with four symmetrically located stamps of a special geometric shape ensuring the joint strength. Subsequently, Kugener [23] analyzed the plastic crimping process for two different wire sizes and two different forming tools while using the Finite Element Method. Likewise, Abbas et al. [24] studied the plastic crimping process joining an electrical connector part and electric wire with the use of FEM. Shirgaokar et al. [25] conducted an analysis of the process clamping a tube around a rod with groove. The studied method of connecting a tube with a rod, without any additional connecting element, can be used in various applications. Researchers conducted parametric tests with the use of FEM in order to study the impact of geometrical parameters on the obtained connection quality.

Wrobel et al. [26] investigated a new crimping method that is designed to form two identical joints using three elements and jaws to deform the material of joining elements. The study case presented in the elaboration was about simultaneously forming two identical joints with the distance of 45 mm between them. Figure 1 shows two joints before the crimping operation.

Due to limited space between two joints and economic assumptions, the joining process in the studied case was based on connecting the two components just by deformation of the material of joining elements without any additional processes. The design of the station assumes the use of eight punches to form the joint; there are two wedges for each forming tool. Mechanical transmissions in the form of two contributing wedges convert vertical motion into horizontal motion of punches. Figure 2 shows the view of the mechanism and two punches with the joint [26].

In Figure 3, joint obtained after the crimping operation performed with the use of two different crimping tools can be seen. Regardless of forming punches, each formed joint has got specific material spills due to the fact that the connecting block flange was not banded all around by the punches during the crimping operation [26].

The researchers compared the that are joints made of the same aluminum alloy, at different main dimensions (diameters), crimped by two different forming tools for each diameters, which was made of the same material and with similar finishing. The conclusions included in the article indicated that the forming process of a smaller joint was more energy consuming than it was in the case of a bigger one. The force that is required to form joints with jaws banding the forming material not all around was reduced in comparison to the second jaw that was used for both diameters of joints.

## 2. Design of the Joint and the Forming Method

The purpose of the study that is presented in this article is to develop a new joint forming method, which has not yet been presented and it is different than the joining process proposed earlier. Furthermore, second purpose is to design a prototype station for forming tight inseparable joints according to a chosen method. In this article, the analysis of the forming method elaborated on the designed joint consisting of two elements is presented: a connecting block and a pipe, which are shown in the Figure 4.

The presented joint has got a similar structure as the previous one; the connecting block has got the flange that will be formed over the flange of the pipe. Additionally, the connecting block has got a prepared place for sealing, e.g., O-ring, which could be used if additional seals are needed to pass the leakage test. The elements of the joint were made of aluminum alloy 6060-T6 using the process of machining, e.g., turning. The mechanical properties and chemical composition of the aluminum alloy can be found in Table 1 and Table 2. Figure 5 shows the described design of the joint.

In the studied case, the forming tool was made in form of a cylindrical part with a recess in its center, consisting of several forming curves. The design use for tools was achievable on account of no restrictions being related to the space that is required for mechanism providing force to the joint being formed, particularly to the connecting block flange. The forming tool, which is made of M261 steel with the use of CNC turning process, deforms the flange of the connecting block and compresses the flange of the pipe, at the same time combining these two elements. The mechanical properties and chemical composition of the material used for the forming tool can be found in Table 3 and Table 4. In the studied joint forming method using the described forming tool, three steps that are similar to the stages of the previous forming method are indicated, which are set out in Figure 6.

## 3. Station Concept

Regardless of the application of the prototype, station several requirements must be met in order to manufacture a proper joint, e.g., the station should be adequately stiff to ensure the repeatability of formed joints; the joint forming area should be well accessible and visible, which is strictly connected to ergonomic and safety issues. Fulfillment of requirements for the station is usually hard to achieve and it is often connected with considering a number of conceptions during the designing process. The selected final design needs to meet the process requirements as well as the budget that is assumed for the project. The forces required for the joint are hard to predict, however understanding the previous case studies allowed for estimating the level of the force at 60 kN. According to the assumption, the station was designed basing on a pneumatic actor with a piston of the size Ø320 mm, which at the pressure of 10 bars can generate the force of 80 kN. Figure 7 provides the described station. In order to collect data during the process, the station was equipped with a force sensor with an amplifier and a displacement sensor, as follows:the force sensor HBM RSCC3/5T with an RM4220 transducer (HBM, Darmstadt, Germany), the nominal range of 5 tons, accuracy of ±0.25% and braking load over 10 tons;a displacement sensor–Keyence GT2-H32 Sn32 (Keyence, Osaka, Japan) with accuracy of 5μm; and,a FESTO pressure sensor supervising its magnitude during the tests.

The data from sensors was transferred to PC with the use of myPCLab transducer connected.

## 4. Methodology of the Experiment

The object of the study is the analysis of a joint forming method regarding the designed joint. Six connecting blocks and six corresponding pipes have been manufactured for the experiment. While considering the prepared components, there can be three specified test cases, which differ in the height of the connecting block flange. Table 5 shows most important dimensions of the sample sets. The described dimensions can be found in Figure 8.

Obtaining repeatable and reliable data during the process of forming the joint with the use of produced components is connected with many different factors, e.g., ensuring similar environmental conditions in the forming process or similar distances between the surfaces of the components. In order to meet the second requirement, the elements of sample sets were manufactured with high tolerances and precision. Unfortunately, in the case of mass production, such accuracy will be hard to achieve and not profitable concerning the economic approach [27].

The joints in the studied forming process were made with the use of one forming tool for each test case, which was possible to obtain due to the appropriate design of components and the forming tool. Sample sets that differed in height of the connecting block flange allowed for changing the point of bending without changing the forming curves of the tool. That clearly affects plastic deformation and compression of the flange of the connecting block and the flange of the pipe, which entails a change in features of the formed joint. In order to study the differences in test cases and the forming process, the obtained sample sets were tested in one air tightness test, two destructive tests, one tensile strength test, and the micrograph analysis of cross-sectioned samples (Table 6).

The tensile strength test was performed on the same prototype station, in which the joints had been formed. The air tightness test was performed using the ATEQ F520 device. The tests were reliable and easy to carry out, because the joint components had been prepared for testing at the designing stage. While designing the connecting block and the pipe, the possibility of using internal threaded connection was taken into account. The obtained data will allow for understanding the cold-forming process and learning the level of energy consumption that was observed while forming the designed joints.

## 5. Results

The first test was based on a micrograph photo of the formed joint cross-section, which enabled us to analyze the process of forming the joint and interference between the materials of the connection block flange and the pipe flange. Thanks to the test, it was possible to determine that key parameters of the proper forming process are: the alpha angle between the connecting block flange and pipe flange, and the point of bending, which can be found in Figure 9.

The value of the first parameter in the case A of sample sets was 27.5°, in cases B & C, the value of the same parameter was 36.5°. The difference in the angle parameters was related to a different bending point and different height of the connecting block flange. In case A, due to the height of the connecting block flange the point of bending was closer to the upper surface of the pipe flange, therefore the material could bend closer to the pipe flange. In the cases B & C, due to lower position of the bending point, the flange of pipe was more tightly compressed. Despite using one forming tool for each case, the resulted outer radius of the formed joint was different in cases B & C and A. It was also considered to have a relation with the point of bending. In the test case A, the formed radius was 2.7 mm, and in the test cases B & C, the formed radius was 3.3 mm. The described parameters for each case can be found in Figure 10.

The basis of the study was the data obtained by sensors mounted on the prototype station while performing the forming process. During the process reaction forces appearing on joints were observed by an HBM force sensor and a Keyence displacement sensor observed the distance that was traveled by the forming tool. The values observed with both sensors for samples sets no. 1, 3, 5 can be found in Figure 11, Figure 12 and Figure 13.

The analysis of the data presented in Figure 11, Figure 12 and Figure 13 determines the maximum forming forces in each case. For the sample set no. 1, it was on the level of 70 kN, for the sample set no. 3, it was on the level of 64 kN, and for the sample set no. 5, it was on the level of 60 kN. The observed reaction forces during the joint forming process for samples sets no. 1, 3, and 5 can be found in Figure 14.

The data presented in Figure 14 shows that for each sample reaction force during forming process is quite regular and with a steady slope. The only gentle decrease of forces is observed before reaching the maximum point on the graph. This different slope on graph can be identified as an increase of energy consumption of process.

On the basis of the analysis, the relationship between the force needed to form the joint and the height of the connecting block flange, and the alpha angle can be observed. The force that is required to form the joint is smaller for joints in the case of samples with a smaller height of the connecting block flange, which can be connected with plastic deformation and compression of material of connecting block flange and can be connected with the alpha angle, which determines the degree of inflection of the connection block flange towards the pipe flange.

The first step of the analysis was the air leak test that was performed with the formed joints due to non-destructive nature of the test. Using the ATEQ F520 device, the sample sets were subjected to the pressure of 0.6 MPa several times. The level of leakage that had been set as faulty was 4 [$\frac{\mathrm{Pa}\cdot L}{s}$]. Table 7 presents the test results. None of the tested sample sets exceeded the established leakage limit.

The last test was the tensile strength test, which was carried out with sample sets no. 1, 3, and 5. The tensile strength test was conducted on prototype station, where the force applied to the joint was generated by ball screw manually powered using a crank. Nevertheless, data acquisition was obtained through a HBM force sensor and a Keyence displacement sensor. The samples were fixed with use of internal threads in the connecting block and the pipe. The connecting block was connected directly to HBM sensor, through special threaded adapter. The pipe was coupled to subassembly of the station, which applied the force to the joint. Tests that were performed on the station were carried out on the basis of standards EN ISO 12996:2013, EN 2591-417:2007, and PN-EN 1993-1-8. The results can be found in Figure 15, Figure 16 and Figure 17.

On the basis of data presented in Figure 15, Figure 16 and Figure 17, the maximum destructive forces can be determined in each case: for the sample set no. 1 the force was at 28 kN, for the sample sets no. 3 and 5 it was at 26 kN. Each value observed with force sensors during tensile test for samples sets no. 1, 3, and 5 can be found in Figure 18.

Data presented in Figure 18 shows that, for each sample, destructive force during the tensile strength test is regular and with a steady slope until the maximum point on the graph. Owing to the fact that forces during the test was manually provided, on the graph the difference in growth rates of tensile forces between sample set no. 1, 3, and 5 can be observed. This fact may affect the measured values of destructive forces, due to the changing dynamics of tests. However, similarity of the slopes and differences of achieved values proves that the obtained results can be compared. The divergent rates can be either connected with different tensile endurance of formed joints. Sample set no. 3 and 5 had similar maximum destructive forces to samples growing rates. Whereas, sample set no. 1 had the biggest destructive force among others, so the growth rate is more leaning than for other samples.

The obtained results allowed for determining the relationship between the force that is needed to destroy the formed connection between elements and to observe the alpha angle. The maximum force that is required to destroy the joint is smaller in the case of joints with a bigger alpha angle. Figure 19 shows the same joint in three different situations: before the forming process, after the forming process, and after the tensile strength test.

According to the figure that is presented above, the destruction of the joint can only be noticed in the case of the pipe element. While testing the flange, the pipe was cut down by the formed flange of the connecting block. The phenomenon of cutting the pipe flange by formed flange of connecting block, rather than opening of formed flange of connecting block, can be clarified by occurring plastic deformation in material of connecting block flange. Assuredly, the work-hardened region occurred in the internal side of flange. Authors attempted to verify this statement, measuring the hardness of the material. However, due to the curved surface and the required depression after the forming process, the measurement cannot be conducted on the internal side of the connecting block flange, only on the outside side. Material hardness measurement has been carried on laboratory Brinell hardness tester KABiD-PRESS B3. Material hardness measurement on not formed surface provided the result of 58 HB, while the obtained hardness on the formed surface was 40 HB. Preformed studies confirmed that the area of necking was located on the external side of connecting block flange, while the work-hardened area may be located on the internal side.

## 6. FEA Simulation

Experiments of cold-formed rounded connections with material deformation of joining elements are expensive due to the requirement of manufacturing sample sets with high tolerances, different forming tools, and time consumption. In order to overcome these difficulties, many researchers decide to simulate the process using different techniques [28]. One of the methods to predict the results of the forming process can be finding equations that are used to predict the mechanical properties of joints. However, the complexity of the joint geometry, three-dimensional nature, and non-linear behavior of the material make it very complicated to elaborate the mathematical formulas. In spite of this, several clinched joints were represented by equations in limited cases [29,30].

The second method that is used to predict results of the forming process can be the Finite Element Analysis. The FEA allows for determining the behavior of almost every formed joint under various loads and geometrical conditions [29,30]. This method was used by other researchers to specify the most advantageous shapes of forming tools in particular cases and to improve the robustness of the forming process [30]. The FEA can overcome several difficulties that are connected with non-linear nature of the process, such as large deformations, material plasticity, and contact interactions [28]. In this study, the Finite Element Analysis was performed in test case A with the use of industrial software Solidworks Simulation Premium.

While investigating the three-dimensional nature of the designed joint with the forming tool that is presented in this study, it was found out that the elements have rotational symmetry along the axis of the tube. Due to the fact mentioned above, it is reasonable to use an infinitely thin slice of half of the designed joint with the forming tool for the FE analysis, which will allow for significantly simplifying the calculations. Figure 20 shows three phases of preparing a model for the FEA simulation.

On the basis of obtained information regarding the forming process, the contact interactions between elements and fixation of elements were determined. In Figure 21, blue and red arrows represent the fixation of the connection block. The axial displacement is blocked along the edge that is marked with red arrows, whereas the displacements in the radial direction are blocked along the edge that is marked with blue arrows. On the second and the third slice of the joint presented in Figure 21, there is marked the area of contact interaction between the connecting block and the pipe that is defined as bonded. On the last slice of the joint presented in Figure 21 there is marked the area of contact interaction between the connecting block and the forming tool defined as no penetration, which prevents two parts from interfering and allows the forming tool to interact with the connecting block. External loads was the next parameter that should have been chosen for the analysis. In the studied case in order not to limit the force applied to the joint, as it was during conducted experiments, external load has been defined as displacement of forming tool. In Figure 21, the green arrows represent the applied displacement in the vertical direction and the sense of its vector—tool was only able to move along the axis; other displacements were forbidden (radial and rotational). In the analysis, the model of forming tool was defined as rigid, when the parts of sample set were defined as deformable.

According to the size of the model and the size of contact area between the forming tool and sample set, the appropriate size of the finite element was determined. The size of finite element was between 0.66 mm and 0.033 mm and the model mesh was made with 1895 elements with 4112 vertices. The type of finite elements used for analysis were shell, circular symmetrical triangular of the second order, made of six vertices (3 corners + 3 middle ones) and three parabolic edges. The distribution of displacements inside the elements is described by quadratic function (second order) and the distribution of strains/stresses by the linear functions. Contact between elements was defined as nod to nod. Figure 22 shows the mesh.

The next parameter of FEA simulation was the coefficient of friction, which was determined on level of 0.5, which is commonly used value for the dynamic coefficient of friction for material pair dry steel and dry aluminum. In the simulation Huber-Mises-Hencky elastic-plastic material model Huber-Mises-Hencky was used. The basis of the joint forming process is the plastic deformation of the material of connecting block elements, which excludes the use of a stress-strain curve for the connecting block material that was obtained in standard tensile tests. The stress-strain curve for aluminum alloy 6060-T6 has to be extrapolated. In the study, the Bridgman approximation was used to determine the maximum stress for the material and the true stress-strain curve. The method used to extrapolate the curve potently affects the results that were obtained by numerical simulation [29]. For the forming tool, a standard stress–strain curve for material has been used. In order to verify the result of FEA simulation, the micrograph cross-section of the clinched joint was compared with the numerically simulated formed joint. The comparison that is shown in Figure 23 proves the correctness of the simulations; although minor differences between the actual and simulated formed joints can be noticed. They were marked in the figure.

The obtained simulation data of the forming process is illustrated in Figure 24, Figure 25 and Figure 26.

According to FE Analysis, the maximum stresses obtained for the simulation (142.7 MPa) did not exceed the determined maximum stresses for the material (202 MPa), therefore fracture criterion was not implemented in simulation. The reaction force resulted in FEA, which was rising until the last step of the analysis, is about 55.7 kN. Confronting the force that is obtained in simulation to data received in experiments indicates the disparity between the forces of 14.3 kN with a predominance of the real test force. The ratio between the forces is 125%. Discrepancy in obtained data can be clarified by differences in the interference between the upper surface of the pipe flange and the surface of the connecting block flange shown in Figure 23. The micrograph section shows interference between two joining elements, which causes the partial compression of the pipe flange material. This may result in increased strength being required to form a joint. The second reason of discrepancies occurred in the obtained data may be a partial transfer of the radial forces arising in the connection that formed as the result of the experiment due to the fact that the formed joint is non-axially loaded or that the joint and the force sensor are non-axially fixed.

## 7. Conclusions

The analysis of data obtained in the forming process and while testing also allows for providing the following statements:Forming the joint in case A was the most energy-consuming due to the highest connecting block flange and the smallest alpha angle of all test cases. The difference in forces between the cases A&B was 6 kN, between the cases A&C it was 10 kN and between the cases B&C-4 kN. The smaller difference between the cases B&C can be related to the similar alpha angle.A relation between the height of the connecting block flange and the alpha angle can be observed.The most durable joint of all test cases was in the case A. The difference between the case A and the cases B & C was about 2 kN. The ratio between them is 107.7%.Each of the test cases passed the leak test. The leakage value did not exceed 4 [(Pa·L)/s]. The tested forming process can be treated as a final operation. It creates the possibility to use the forming process in a situation when the brazing or welding processes cannot be used.The reaction force that was obtained in the FE Analysis was lower than the reaction force included in the experiment data. The ratio between the forces was 125%. The difference can be clarified by a lack of interference between the upper surface of the pipe flange and the surface of the connecting block flange or by transferring the radial forces (arising as the result of the experiment) to the sensor due to the fact that the formed joint is non-axially loaded or that the joint and the force sensor are non-axially fixed.Due to obtained high tensile strength of formed joints, further research should be undertaken to improve the cost-effectiveness of the process. Efforts should be made to reduce the energy that is needed to form the joint, even if it reduces the strength of the joint.Further research should consider the thickness of the connecting block flange, due to the fact that, during the test, the pipe flange was cut down by the formed flange of the connecting block without any damage of the connecting block flange. The connecting block flange should be as high as in the case A due to better resistance of test case to ambient conditions (there is the smallest gap between the pipe and the connecting block). A reduction of thickness of the connection block flange should result in lower forming forces and less wearing of the forming tool.Further research could either consider changing forming shape of the tool. The change of forming curve may result in a reduction of the alpha angle, which may affect the tensile strength and the reduction of leakage. The analysis should be conducted with use of the FE method, which can reduce the costs that are associated with the production of various types of tools. The FE analysis of the formed joints should be extended with the analysis of the tensile strength of the formed joints. The designed shape that will be considered the most advantageous should be experimentally analyzed to establish whether the leakage was reduced or not.

## Figures and Tables

**Figure 1 materials-12-01061-f001:**
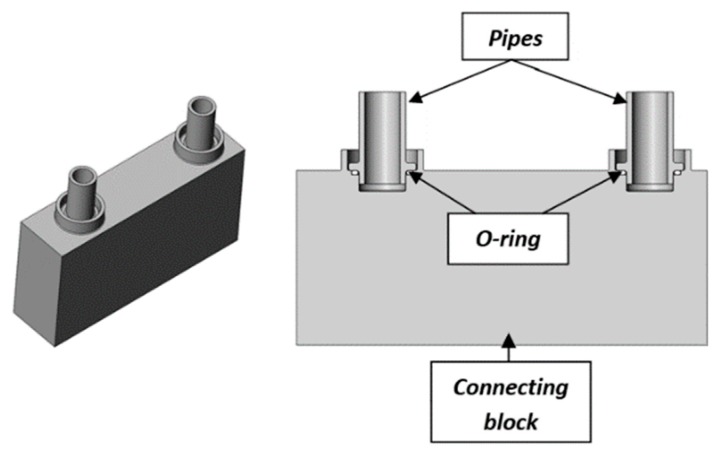
Cross-section and three-dimensional (3D) view of two joints, consisting of two pipes and one connecting block, before the crimping process carried out in one simultaneous forming operation for two identical joints. [26].

**Figure 2 materials-12-01061-f002:**
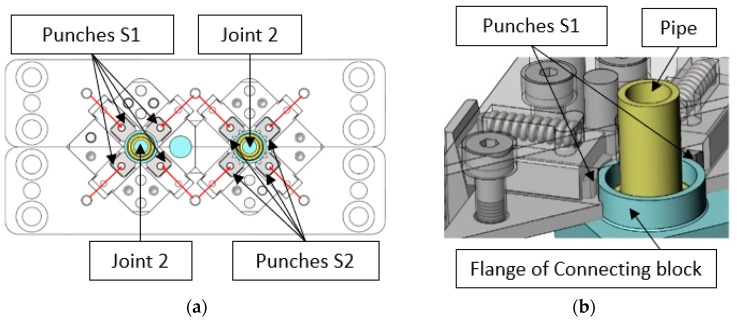
Mechanism used for forming two joints simultaneously: (**a**) the top view of mechanism with marked directions of movements, joints and forming punches of two different shapes S1 and S2; and, (**b**) zoomed 3D view of two punches with the joint being in contact with the flange of the connecting block [26].

**Figure 3 materials-12-01061-f003:**
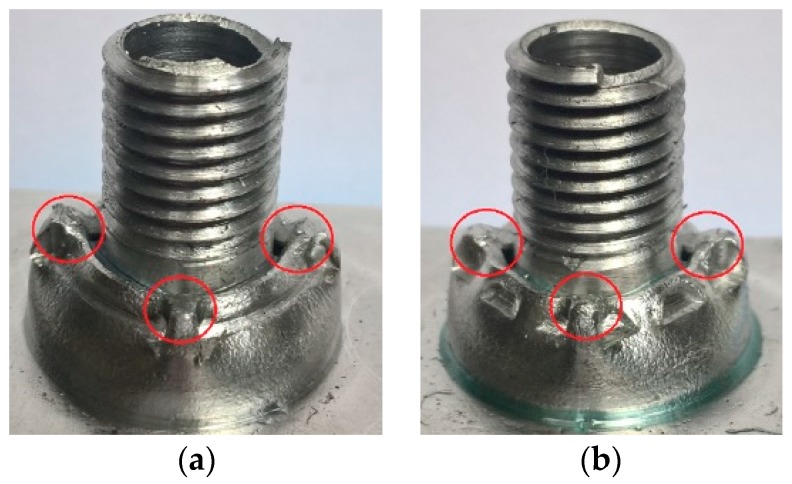
Formed joints with marked characteristic material spills: (**a**) the joint after crimping with the use of Tool S1; (**b**) the joint after crimping with the use of Tool S2 [26].

**Figure 4 materials-12-01061-f004:**
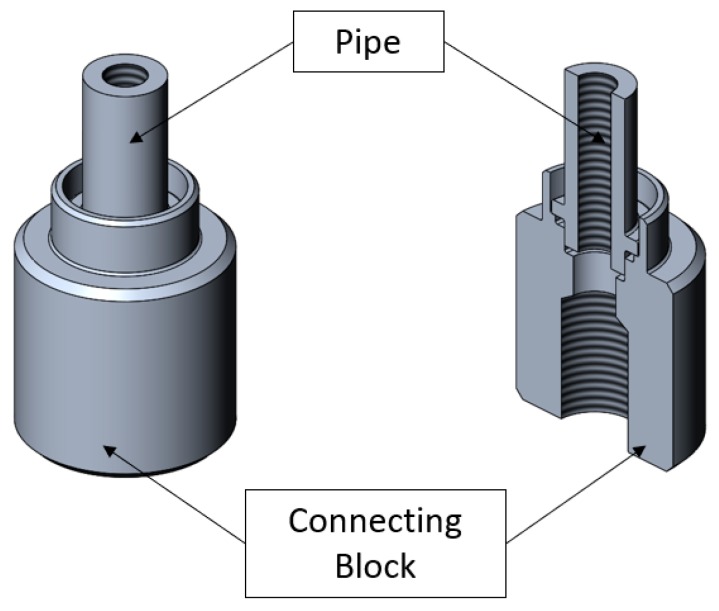
Axonometric view and cross-sectional axonometric view of the designed elements: a pipe and a connecting block to be joined.

**Figure 5 materials-12-01061-f005:**
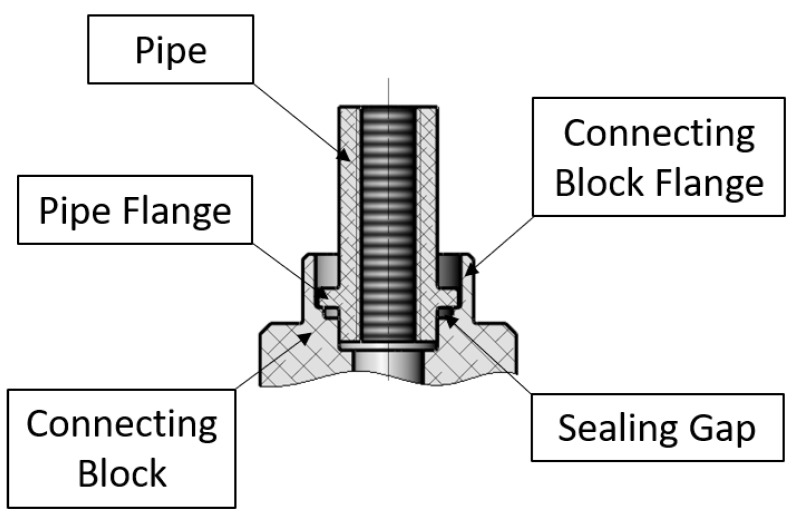
Cross-section view of elements to be joined, with labelled consisting elements, i.e., a pipe and a connecting block, and significant areas of the joint, i.e., a pipe flange, a connecting block flange and a sealing gap.

**Figure 6 materials-12-01061-f006:**
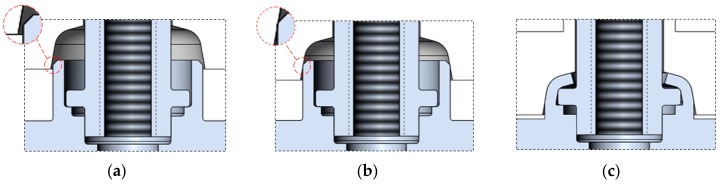
Cross-sections of the joint in three different forming steps: (**a**) without contact with the forming tool (the contact area can be seen in the excerpt in the upper left-hand corner of the figure); (**b**) in contact with the forming tool (the contact area can be seen in the excerpt in the upper left-hand corner of the figure); and (**c**) while shaping by the forming tool.

**Figure 7 materials-12-01061-f007:**
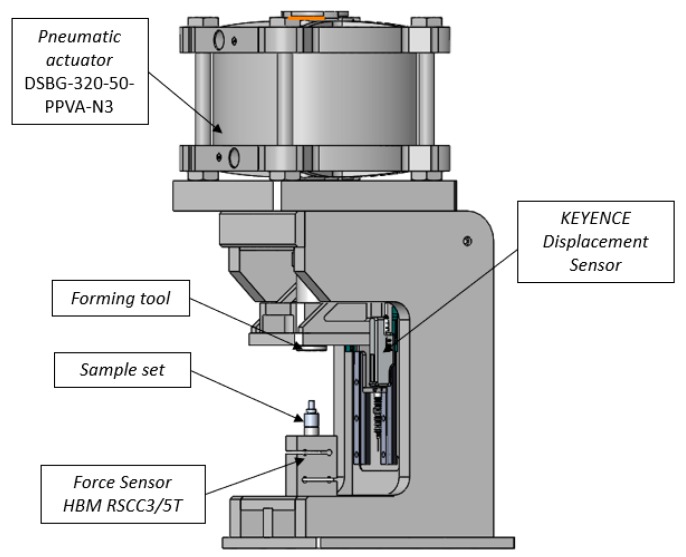
The prototype station designed to form cold-formed rounded joints with labelled components: a pneumatic actuator, a displacement sensor, a joint to be formed, a forming tool, and a force sensor.

**Figure 8 materials-12-01061-f008:**
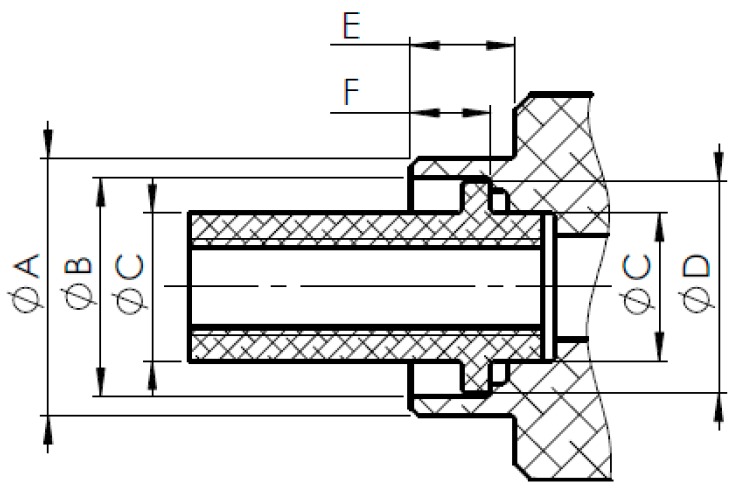
Cross-section of the sample set with important dimensions.

**Figure 9 materials-12-01061-f009:**
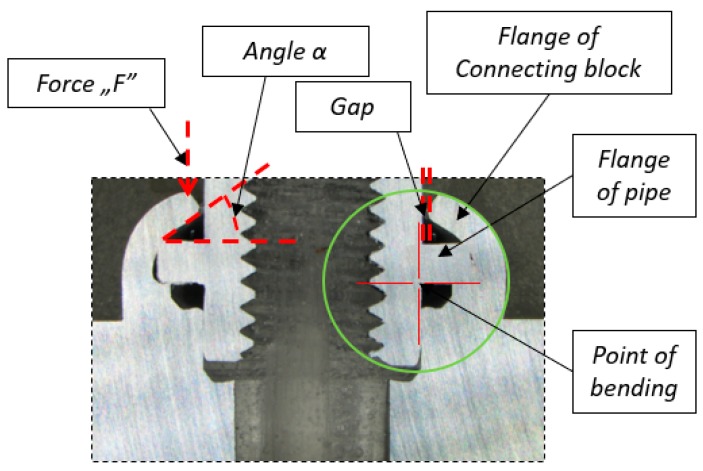
Micrograph section of the formed joint with significant areas of the pipe and the connecting block elements labelled and significant process parameters marked in red (the point of bending, the angle).

**Figure 10 materials-12-01061-f010:**
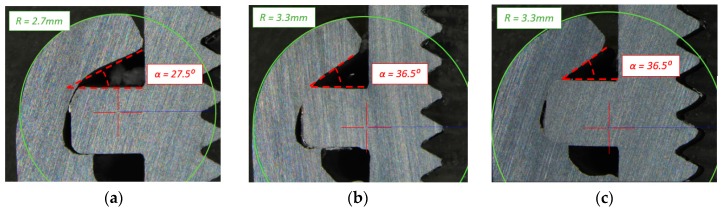
Micrograph section of formed joints with the significant process parameters marked in red: the point of bending and the alpha angle between the flange of the connecting block and the flange of the pipe for: (**a**) Case A; (**b**) Case B; and (**c**) Case C.

**Figure 11 materials-12-01061-f011:**
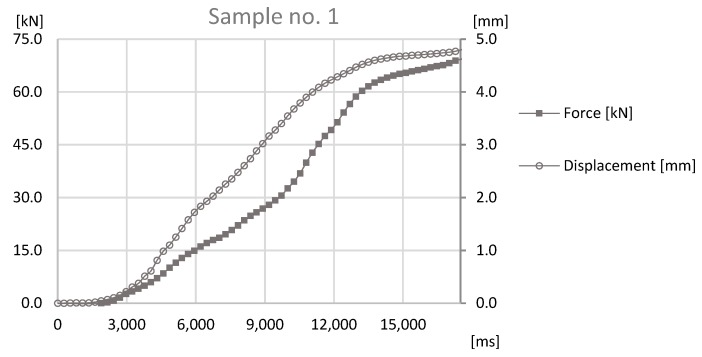
The graph of force-at-time and displacement-at-time during the joint forming process for Sample no. 1.

**Figure 12 materials-12-01061-f012:**
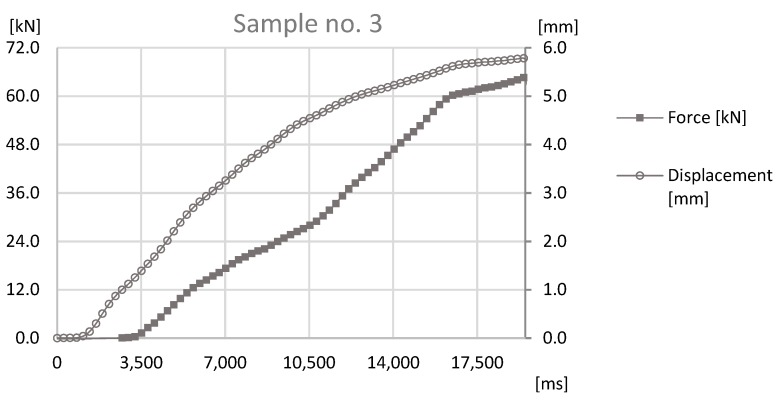
The graph of force-at-time and displacement-at-time during the joint forming process for Sample no. 3.

**Figure 13 materials-12-01061-f013:**
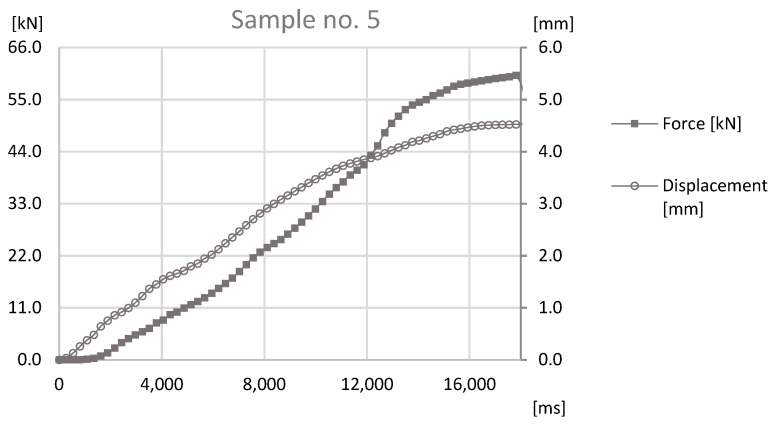
The graph of force-at-time and displacement-at-time during the joint forming process for Sample no. 5.

**Figure 14 materials-12-01061-f014:**
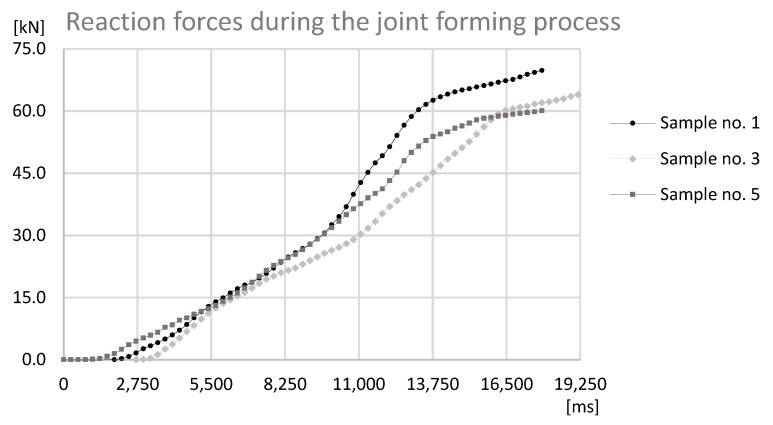
The graph of reaction force-at-time during the joint forming process for Sample no. 1, 3, and 5.

**Figure 15 materials-12-01061-f015:**
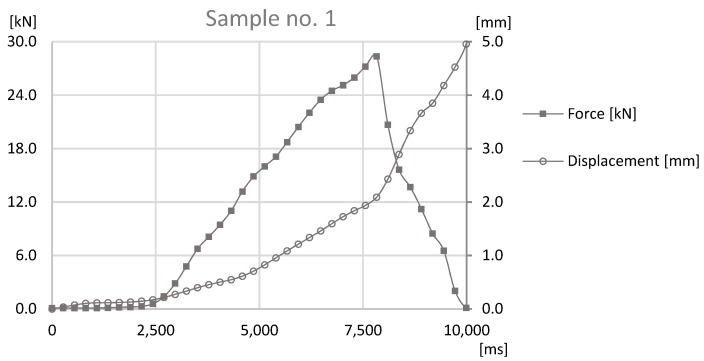
The graph of the force-at-time and displacement-at-time for the tensile strength test for Sample no. 1.

**Figure 16 materials-12-01061-f016:**
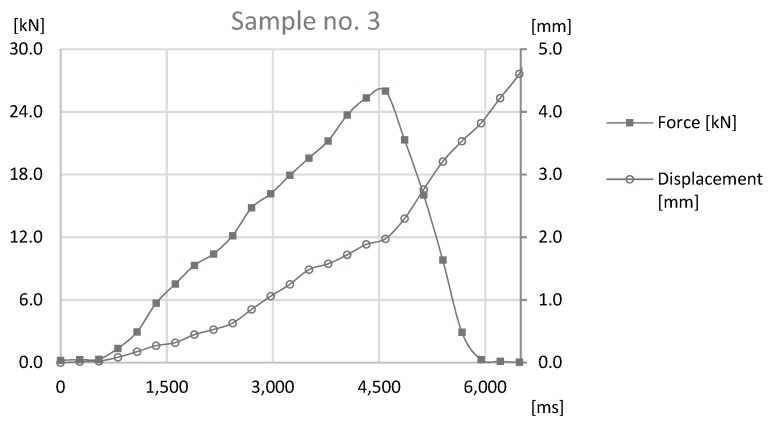
The graph of the force-at-time and displacement-at-time for the tensile strength test for Sample no. 3.

**Figure 17 materials-12-01061-f017:**
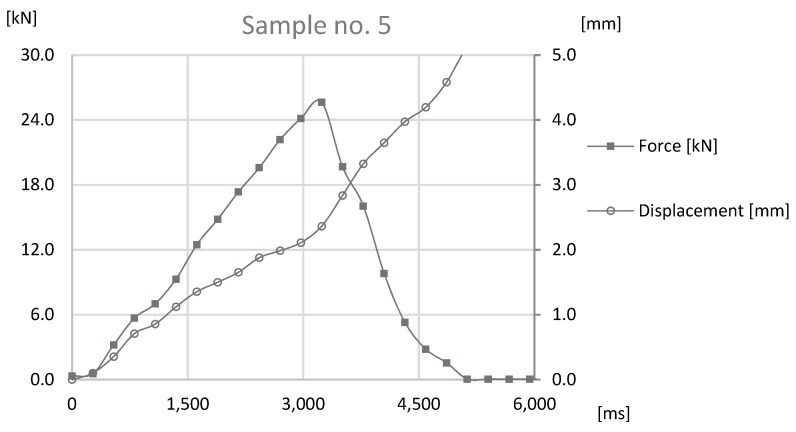
The graph of the force-at-time and displacement-at-time for the tensile strength test for Sample no. 5.

**Figure 18 materials-12-01061-f018:**
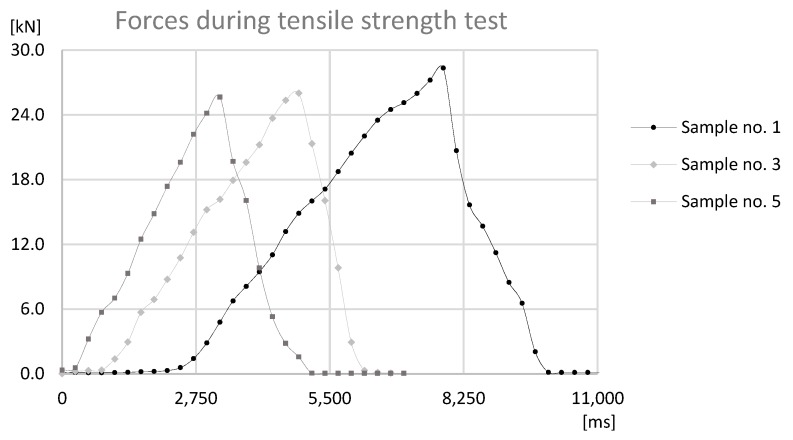
The graph of destructive force-at-time during the tensile strength test for Sample no. 1, 3, and 5.

**Figure 19 materials-12-01061-f019:**
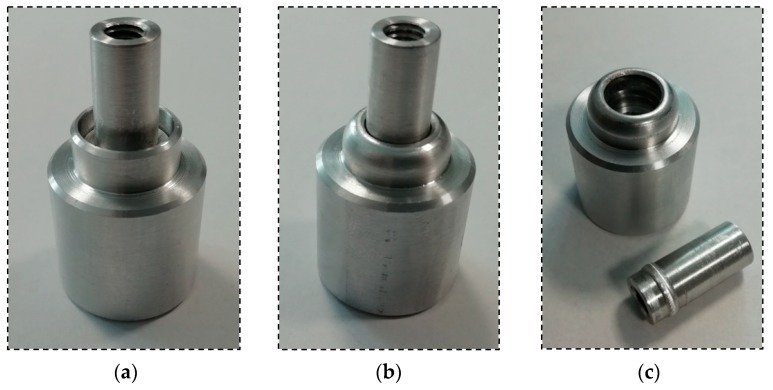
The designed joint: (**a**) before the forming process; (**b**); after the forming process; and, (**c**) after the tensile strength test.

**Figure 20 materials-12-01061-f020:**
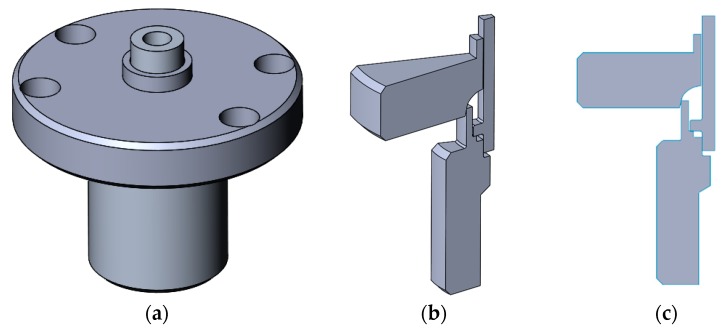
Simplification of the FEA model: (**a**) the designed joint with the forming tool; (**b**) 3D simplification of the joint with the forming tool using rotational symmetry; (**c**) two-dimensional (2D) simplification of the joint with the forming tool using rotational symmetry.

**Figure 21 materials-12-01061-f021:**
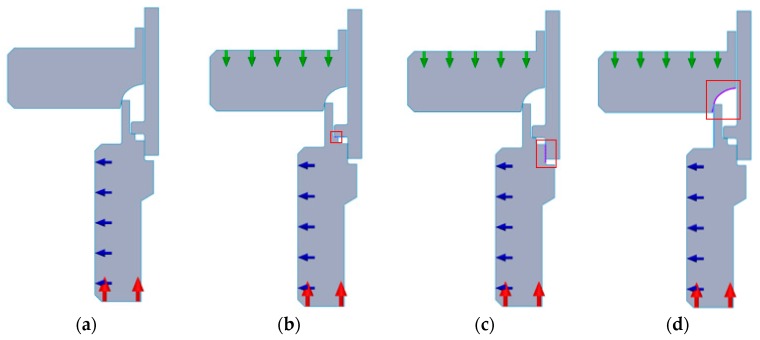
Fixation and contact interactions of the FEA model: (**a**) fixation of the connection block marked with blue and red arrows; (**b**) the area of contact interaction between the connecting block and the pipe along the axis of the joint; (**c**) the area of contact interaction between the connecting block and the pipe in the radial direction of the joint; and, (**d**) the area of contact interaction between the connecting block and the forming tool.

**Figure 22 materials-12-01061-f022:**
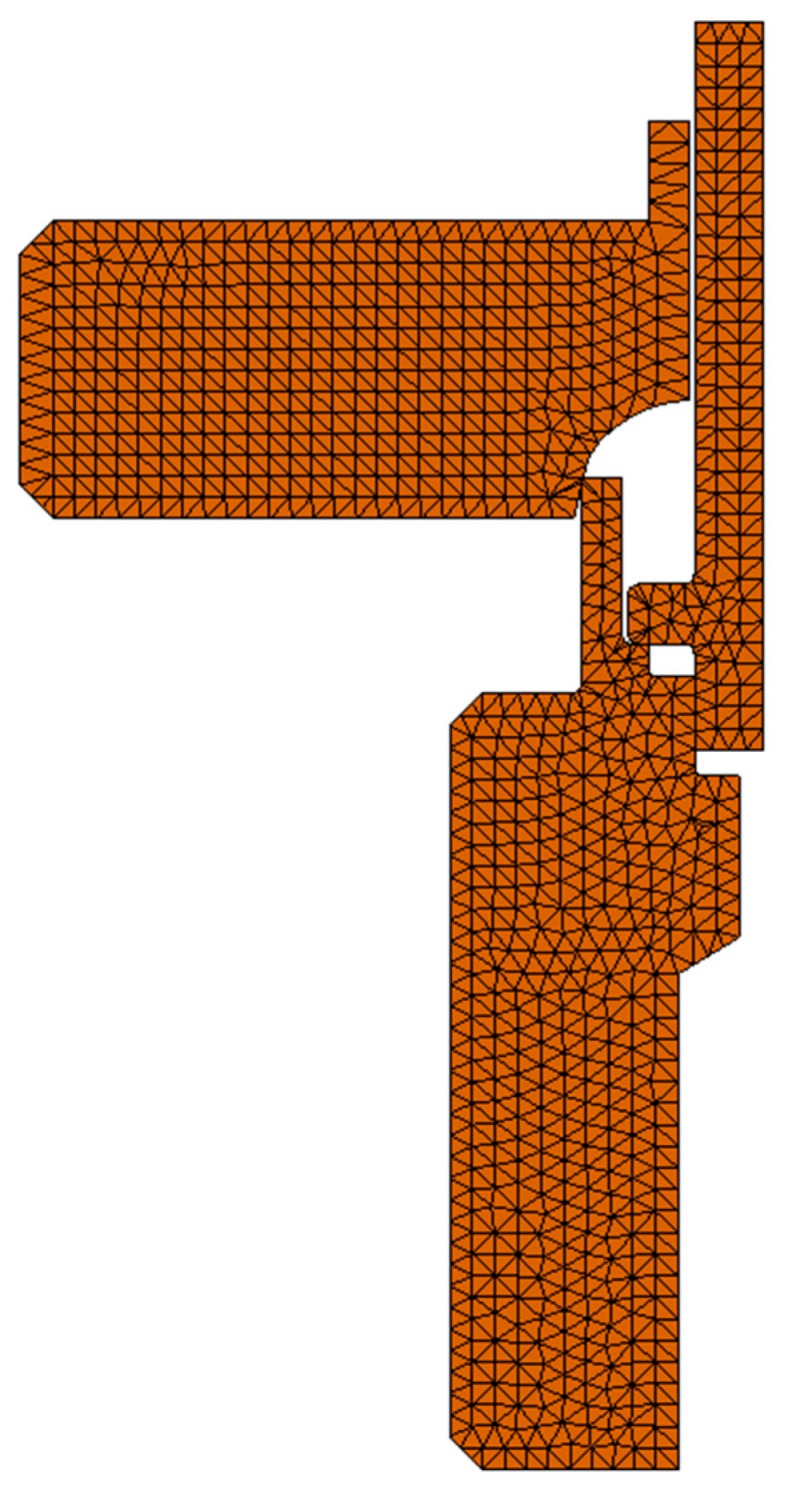
Simplified FEA model of the joint with the forming tool with the mesh.

**Figure 23 materials-12-01061-f023:**
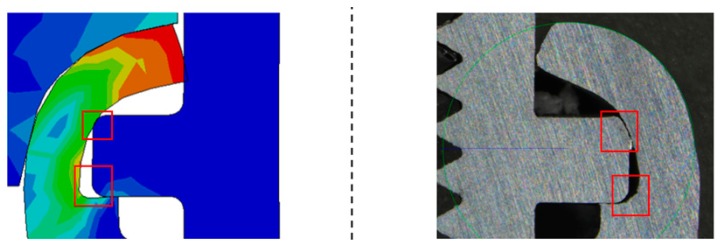
Comparison of the FEA simulation model and the micrograph section with minor differences between the actual and simulated formed joints marked in red.

**Figure 24 materials-12-01061-f024:**
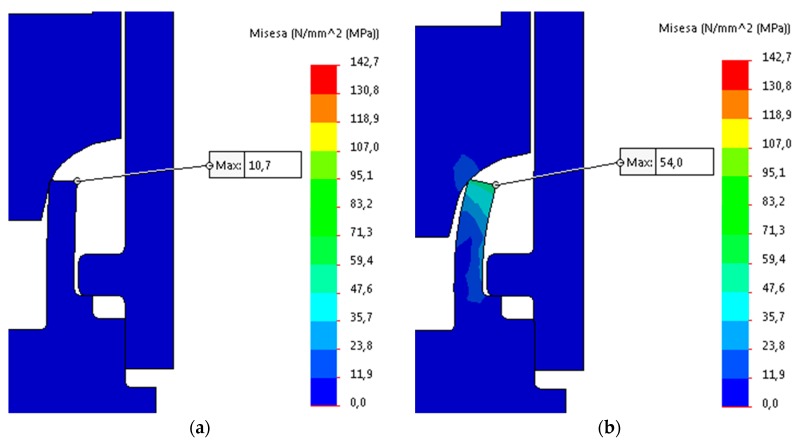
FEA simulation results: (**a**) step 1 (t = 0.1 s); and, (**b**) step 2 (t = 0.25 s).

**Figure 25 materials-12-01061-f025:**
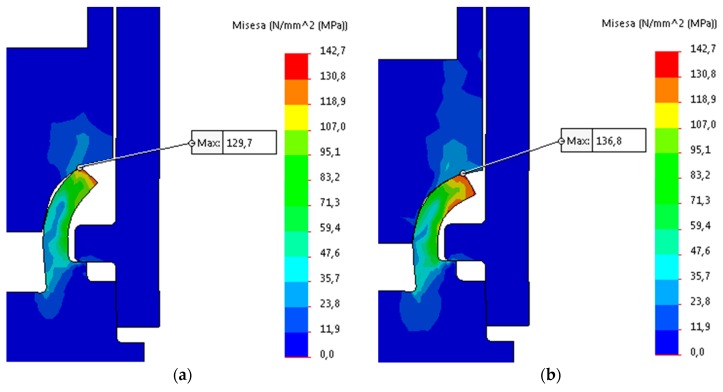
FEA simulation results: (**a**) step 3 (t = 0.48 s); and, (**b**) step 4 (t = 0.6 s).

**Figure 26 materials-12-01061-f026:**
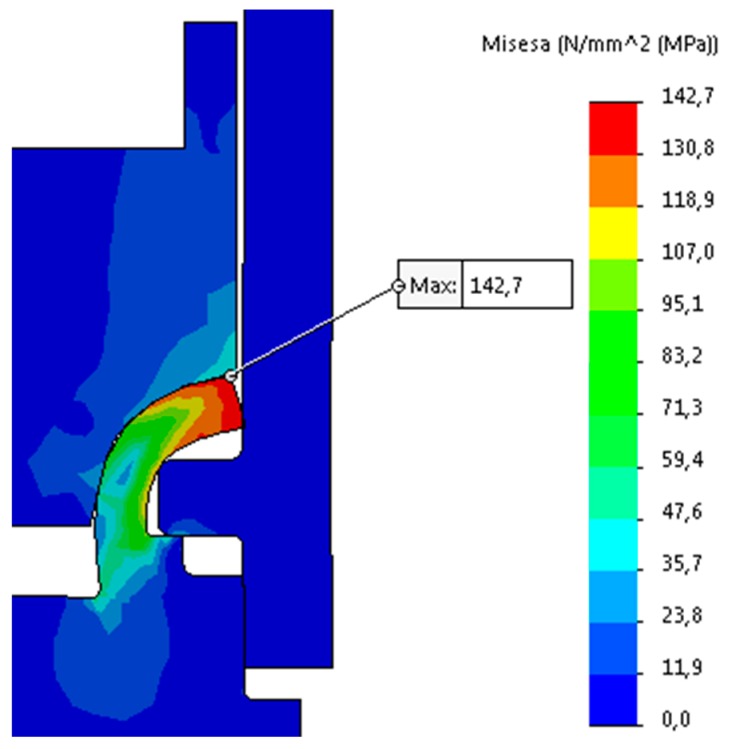
FEA simulation results, final step 17 (t = 0.73 s).

**Table 1 materials-12-01061-t001:** Chemical composition of alloy 6060-T6.

Alloy	Si (%)	Fe (%)	Cu (%)	Mn (%)	Mg (%)	Cr (%)	Zn (%)	Ti (%)	Al (%)
6060-T6	0.30–0.60	0.10–0.30	max. 0.10	max. 0.10	0.35–0.60	max. 0.05	max. 0.15	max. 0.10	Balance

**Table 2 materials-12-01061-t002:** Mechanical properties of alloy 6060-T6.

Alloy	Yield Strength (MPa)	Ultimate Tensile Strength (MPa)	Hardness (HB)
6060-T6	140	170	60

**Table 3 materials-12-01061-t003:** Chemical composition of alloy M261.

Alloy	C (%)	Si (%)	Mn (%)	Cr (%)	Ni (%)	Cu (%)	Al (%)	Fe (%)
M261	0.13	0.30	2.00	0.35	3.5	1.2	1.2	Balance

**Table 4 materials-12-01061-t004:** Mechanical properties of alloy M261.

Alloy	Yield Strength (MPa)	Ultimate Tensile Strength (MPa)	Hardness (HRc)
M261	1180	1250	38–42

**Table 5 materials-12-01061-t005:** Dimensional description of sample sets (including tolerances) regarding particular sample cases.

No.	Case	ØA (mm)	ØB (mm)	ØC (mm)	ØD (mm)	E (mm)	F (mm)
1	A	Ø16 h6	Ø13.6 H7	Ø9.2 H7/h7	Ø13.2 h7	6.5	$5_{+0}^{+0,05}$
2	B	Ø16 h6	Ø13.6 H7	Ø9.2 H7/h7	Ø13.2 h7	6	$4.5_{+0}^{+0,05}$
3	C	Ø16 h6	Ø13.6 H7	Ø9.2 H7/h7	Ø13.2 h7	5.5	$4_{+0}^{+0,05}$

**Table 6 materials-12-01061-t006:** Detailed division of sample sets regarding tests and particular cases.

No.	Case	Stage of Connection Technology	Test 1 Micrography	Test 2 Leak Test	Test 3 Tensile Strength	Sealing
1	A	Final		X	X	no O-ring
2	A	Final	X	X		no O-ring
3	B	Final		X	X	no O-ring
4	B	Final	X	X		no O-ring
5	C	Final		X	X	no O-ring
6	C	Final	X	X		no O-ring

**Table 7 materials-12-01061-t007:** Detailed results of non-destructive air leak test regarding sample cases.

No.	Case	Pressure (MPa)	Leakage Pa·L/s
1	A	0.6	3
2	A	0.6	3
3	B	0.6	3
4	B	0.6	2
5	C	0.6	2
6	C	0.6	3

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
