# Peer review of "Designing and Testing Cold-Formed Rounded Connections Made on a Prototype Station"

_materials, 2019, doi:10.3390/ma12071061_

Reviewer 1 Report

The aim of the manuscript is to present experimental and simulated results of tube-to-block cold round formed connections. The significant contribution of the research is the comprehensive and supportive design-data for a novel cold round formed connection.

The strength of this manuscript is the comprehensive nature of the design study. Authors attempted all avenues to show the mechanical robustness and applicability of the design.

The comprehensive experimental study shows a possible scope of further improvement of the tensile strength and reduction of leakage by providing a conformable radius over the tube-flange according to the forming tool radius. This will possibly flow the concentrated induced stress through the flanges during the forming process and will reduce the possibility of easy failure of the tube-flange under reverse loading during the tensile test. The conformed radius of the tube-flange will eliminate alpha-angle, allow more compact seating position for the block-flange and will possibly reduce the leakage. In this case, the simulation of the tensile test could be more informative.

Authors are advised to include such thoughtful discussion about the scope of future improvement in the Conclusion Section.

Line 326: What is the basis of the leakage limit (4 Pa.l/s)? Any reference to the standard of practice on this scope is required.

Figure 15-17 indicate the rate of testing was not consistent for all three cases. That could be the reason for showing different tensile strengths for the three cases. Please explain.

Author Response

First of all, the authors are very grateful to Editor for quick and professional review process and for the Reviewers for careful reading of the paper and suggestions that allowed improving its quality. All issues raised by the reviewers were considered. A revision of the article has been performed in order to improve its effectiveness and remove errors. The changes introduced in the manuscript were indicated with a colour. All actions are listed below:

1. What is the basis of the leakage limit (4 Pa.l/s)? Any reference to the standard of practice on this scope is required.

Answer: Thank you for a question, the leakage limit determined empirically by automotive industry reference has to this has been added.

2. Indicate the rate of testing was not consistent for all three cases. That could be the reason for showing different tensile strengths for the three cases. Please explain.

Answer: Thank you for question, explanation has been added.

3. The comprehensive experimental study shows a possible scope of further improvement of the tensile strength and reduction of leakage by providing a conformable radius over the tube-flange according to the forming tool radius. This will possibly flow the concentrated induced stress through the flanges during the forming process and will reduce the possibility of easy failure of the tube-flange under reverse loading during the tensile test. The conformed radius of the tube-flange will eliminate alpha-angle, allow more compact seating position for the block-flange and will possibly reduce the leakage. In this case, the simulation of the tensile test could be more informative. Authors are advised to include such thoughtful discussion about the scope of future improvement in the Conclusion Section.

Answer: Thank you for advice, additional conclusion point has been added. The discussion about further research including changing the forming curve of forming tool with possible improvements of joint has been added.

Reviewer 2 Report

I think the paper is very interesting for readers of this journal.

Author Response

I think the paper is very interesting for readers of this journal.

Thank you very much for your positive opinions.